# Bta-miR-125a Regulates Milk-Fat Synthesis by Targeting *SAA1* mRNA in Bovine Mammary Epithelial Cells

Xiaogang Cui ⬤, Tianqi Yuan, Zhengyu Fang, Jiao Feng and Changxin Wu *

Key Laboratory of Medical Molecular Cell Biology of Shanxi Province, Institute of Biomedical Sciences, Shanxi University, Taiyuan 030006, China; helloxiaogangcui@sxu.edu.cn (X.C.); 202023105015@email.sxu.edu.cn (T.Y.); 202023118003@email.sxu.edu.cn (Z.F.); fengjiao2018@sxu.edu.cn (J.F.)
* Correspondence: cxw20@sxu.edu.cn

**Abstract:** The nutritional value of cow milk mainly depends on its fatty acid content and protein composition. The identification of genes controlling milk production traits and their regulatory mechanisms is particularly important for accelerating genetic progress in the breeding of dairy cows. On the basis of mammary gland transcriptome analyses, in this study we identified an miRNA, bta-miR-125a, that could control bovine milk-fat production by targeting the $3'$ untranslated region (UTR) of the serum amyloid A-1 (SAA1) mRNA. The presence of synthetic bta-miR-125a (i.e., an miR-125a mimic) significantly down-regulated the expression of luciferase from mRNAs containing the binding sequence for bta-miR-125a in the $3'$-UTR in a dual-luciferase reporter assay. Furthermore, the presence of the miR-125a mimic decreased the steady-state level of the *SAA1* protein, but increased the accumulation of triglycerides and cholesterol content in bovine mammary epithelial cells (MAC-Ts). Blocking the function of bta-miR-125a using a specific inhibitor decreased the level of triglycerides and cholesterol content in the cells. These results indicate that bta-miR-125a can serve as a positive regulator of lipid synthesis in mammary epithelial cells, which acts by targeting *SAA1* gene expression.

**Keywords:** mammary epithelial cells; milk-fat synthesis; miR-125a; *SAA1*; MAC-T

## 1. Introduction

Cow milk is regarded as a basic food in many diets worldwide [1], and contains nearly all of the essential elements required for healthy human nutrition. Milk fat is a milk quality indicator, and a determining element of the nutritional value of milk. It is composed of lipid droplets, which mainly consist of triacylglycerides (TAGs), which are synthesized and released from the bovine mammary epithelial cells. The milk-fat levels and composition are affected by various factors, including heredity, nutrition, physiological conditions and the environment. More studies have revealed some miR-NAs affecting lipid metabolism by targeting genes, e.g., ATP binding cassette transporter A1 (*ABCA1*), with Chen et al. reporting that miR-106b functions through ABCA1 by mRNA and protein levels [2]; however, the underlying mechanisms remain poorly understood [3–5].

Serum amyloid A (SAA) is the most prominent and highly up-regulated protein during acute inflammation [6]. The SAA protein is synthesized in the liver, and has been shown to be involved in the metabolism of lipids [7]. Four isoforms of SAA have been identified through amino-acid sequence analysis: *SAA1*, *2*, *3* and *4* [8]. The *SAA1* and *SAA2* isoforms are mainly synthesized in hepatocytes, and are primarily associated with high-density lipoprotein (HDL) [9]. In addition, the expression of *SAA1* and *SAA2* in non-hepatic tissues has also been reported [10]. *SAA1* is involved in the development of the mammary gland through NF-κB-dependent signaling, while the over-expression of *SAA1* may accelerate apoptosis and suppress mammary epithelial cell growth [11]. Furthermore, seven SNPs (single nucleotide polymorphisms) in the *SAA1* gene have been significantly associated

with milk yield and composition traits [12]. Unlike *SAA1* and *SAA2*, *SAA3* isoform is present in a lipid-poor form, not associated with HDL [7]. The *SAA4* isoform is an HDL-associated apolipoprotein, and constitutively expressed at relatively low levels in both human and mouse liver [13]. To date, the SAA4 isoform function is largely unknown [14].

MicroRNAs (miRNAs) are small, non-coding RNAs that regulate gene expression at the post-transcriptional level in various biological processes [15]. It has been estimated that the expression of 30% of protein-coding genes is regulated by miRNAs [16,17].

The mammary gland is a uniquely specialized organ in humans and mammals. The specialized tissues, including the mammary epithelial cells in the mammary gland, undergo proliferation, differentiation and apoptosis. Various roles of miRNAs in the development of the mammary gland in humans [18,19], the maintenance of mammary epithelial progenitor cells in mice [20] and the proliferation and differentiation of mammary epithelial cells in humans [21,22] have been uncovered, as have those in the outgrowth of epithelial ducts in mice [23]. In mammary epithelial cells, several miRNAs have been found to be involved in the production of milk fat by targeting different genes. miR-224 can inhibit the secretion of milk fat by down-regulating the expression of the acyl-coenzyme A dehydrogenase (*ACADM*) and aldehyde dehydrogenase2 (*ALDH2*) genes [24]. The over-expression of miR-224 has been associated with a decrease in triglycerides in mammary epithelial cells [24]. Contrary to miR-224, miR-21-3p promotes triglyceride production in cow mammary epithelial cells—through the inhibition of the elongation of the very long chain fatty acids protein 5 (*Elovl5*) gene, which is a gene important in lipid metabolism—by catalyzing the elongation of fatty acids [25]. Furthermore, miR-15a inhibits the vitality and lactation of mammary epithelial cells by targeting the *GHR* gene [26], which is associated with milk composition. Moreover, miR-27a-3p can inhibit milk-fat synthesis by dairy cow mammary epithelial cells (MAC-Ts) by targeting peroxisome proliferator-activated receptor gamma (PPARG), which enhances the synthesis of monounsaturated fatty acids by controlling stearoyl-CoA desaturase [4].

In our previous studies, we analyzed the mammary gland epithelial tissues of four lactating Holstein cows with extremely high and low milk-protein (PP) and fat percentages (FP), using RNA sequencing (RNA-seq) and small RNA-seq [27,28]. We identified 21 differentially expressed genes as potential targets for some of the 71 differentially expressed miRNAs, including the *SAA1* gene and bta-miR-125a, respectively [29]. Based on those preliminary results, we hypothesized that bta-miR-125a might regulate the expression of the *SAA1* gene. In this study, this hypothesis was tested using different techniques, including dual-luciferase reporter assays, quantitative reverse-transcription PCR (qRT-PCR) and oil red O staining assays, in order to investigate the functional relevance of bta-miR-125a for the production of milk fat in dairy cow mammary epithelial cells (MAC-Ts). Our results shed new light on the network of miRNAs involved in the production of milk.

## 2. Materials and Methods

### 2.1. Cell Cultures

Human HEK293T and bovine MAC-T cells were grown in Dulbecco's modified Eagle's medium (DMEM; Boster, Wuhan, China) containing 10% fetal bovine serum (Gibco, Waltham, MA, USA), 100 IU/mL penicillin and 100 μg/mL streptomycin (Solarbio, Beijing, China) at 37 °C under 5% $CO_2$.

### 2.2. Plasmid Construction and Site-Directed Mutagenesis

The binding sequence of bta-miR-125a was predicted to be in the 3′ untranslated region (UTR) of the *SAA1* mRNA. Thus, the DNA sequence for this region was amplified by PCR, using bovine mammary gland DNAs as templates and a primer pair (p*SAA1* 3′UTR Forward: 5′-GCTGCCTCTCTCTGCTCAG-3′; p*SAA1* 3′UTR Reverse: 5′-TTTTGTTTGACCCAAATATAGTGAGGATAAAGGT-3′; Sangon, Shanghai, China). The PCR conditions were as follows: 94 °C for 5 min, followed by 35 cycles of 94 °C for 30 s; 60 °C for 30 s; and 72 °C for 45 s; and a final extension at 72 °C for 10 min. The 683 bp

PCR products were digested with PmeI and XbaI, and then inserted into the plasmid pmirGLO Dual-Luciferase miRNA Target Expression Vector (PmiRGLO, Promega, Madison, WI, USA), in order to obtain the recombinant plasmid PmiR-*SAA1*–3′UTR-wild type (WT) (Figure 1A). The seed sequences recognized by miR-125a in the plasmid PmiR-*SAA1*–3′UTR-wild type (WT) were deleted using a QuikChange site-directed mutagenesis kit (Stratagene, La Jolla, CA, USA) to generate the plasmid PmiR-*SAA1*–3′UTR-mutant (MUT), which was used as a negative control (Figure 1B). The plasmids were all confirmed by DNA sequencing and used in the luciferase reporter assay.

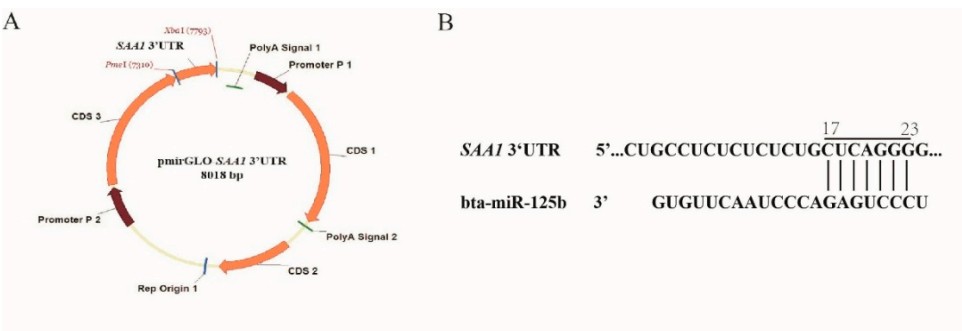

**Figure 1.** (**A**) The pmirGLO vectors with the predicted 3′UTR target sequences of the expressed gene, pmirGLO-SAA1-3′UTR; (**B**) Locations and sequences of the miRNA target sites in the 3′UTR of the expressed gene. The sequences of the miRNAs are indicated, along with mutations introduced in the target sites (underlined nucleotides) for generating the mutated reporter constructs.

### 2.3. Luciferase Reporter Assays

Synthetic bta-miR-125a (miR mimic), bta-miR-125a-specific inhibitor miRNA (miR inhibitor) and non-specific negative control miRNA (NC) were synthesized and purchased from GenePharma (Shanghai, China). Luciferase reporter assays were performed in HEK293T cells. The cells were plated into six-well plates at $5.0 \times 10^5$ cells/well for 24 h. Then, 30 ng of the empty PmiRGLO vector, PmiRGLO-*SAA1*–3′UTR-wild type (WT) or PmiR-*SAA1*–3′UTR-mutant (MUT) was co-transfected with a 30 nM concentration (final concentration) of the miRNA mimic, miRNA inhibitor or NC (GenePharma) in various combinations in each well, using 1 µL of Lipofectamine 3000 (Invitrogen, Waltham, MA, USA). At 24 h post-transfection, the relative activities of firefly luciferase were measured using a TECAN Infinite 200 multifunctional microplate reader (Tecan, Männedorf, Switzerland), and normalized to the relative activity of *Renilla* luciferase. The experiments were performed in triplicate, and the presented data are averages from three independent experiments.

### 2.4. Transient Transfection in MAC-T Cells

MAC-T cells were seeded in 24-well cell culture plates at a density of approximately $4 \times 10^5$ cells per well, with quadruplicate wells per group. For each well, a 50 nM concentration (final concentration) of the miR-125a miRNA mimic, miRNA-inhibitor or NC (GenePharma) was mixed with 1 µL of Lipofectamine 3000 (Invitrogen) in 50 µL of opti-MEM for 15 min at room temperature. Then, the mixture was added into each well. At 24 h, 48 h and 72 h post-transfection, the cells were collected and processed for further analysis.

### 2.5. RNA Isolation and Quantitative RT-PCR (qRT-PCR)

Total RNA was isolated from transfected MAC-T cells using TRIzol reagent (Invitrogen, Carlsbad, CA, USA), following the manufacturer's instructions. The concentration and purity of the total RNA were determined using a Nanodrop 2000 spectrophotometer (Thermo, Waltham, MA, USA). RNA samples with optical density values between 1.8 and 2.0 at 260/280 nm (OD 260/280) were used in the qRT-PCR analysis. The first-strand complementary DNA (cDNA) was synthesized using an miRcutePlus miRNA First-Strand

cDNA Kit (Tiangen, Beijing, China), following the manufacturer's protocol. qRT-PCR was performed using TB Green Premix Ex Taq II (SYBR Green, TAKARA) and a miRcute Plus miRNA qPCR Kit (SYBR Green) (Tiangen, Beijing, China) on a LightCycler 480 II Real-time RT-PCR System (LightCycler, Indianapolis, IN, USA). The levels of the *GAPDH* mRNA or U6 RNA were tested as endogenous controls. The $2^{-\Delta\Delta CT}$ method was applied to calculate the relative expression of the indicated genes. The results are representative of at least three independent experiments. The primers used in the qRT-PCR are listed in Tables 1 and 2.

**Table 1.** Primer sequences for qPCR.

| Name | RT Primer (5′ to 3′) | Forward Primer (5′ to 3′) | Reversed Primer (5′ to 3′) | Tm (°C) |
|---|---|---|---|---|
| bta-miR-125a | CTCAACTGGTGTCGTGGAG TCGGCAATTCAG TTGAG CACAGGTT | GGGCTTCCCTGA GACCCTTT | CTCAACTGGTGTCGTG GAGTC | 60 |
| U6 | TIANGEN: CD201–0145 | | | 60 |

**Table 2.** Primer sequences used for qRT-PCR.

| Gene Name | Forward Primer Sequence | Reverse Primer Sequence | Amplicon (bp) | Tm (°C) |
|---|---|---|---|---|
| *SAA1* | AGTCCACAGCCAGTGGATGT | ATCTCTGAA TATTTTCTCTGGCATC | 2433 | 60 |
| *GAPDH* | AGATGGTGAAGGTCGGAGTG | CGTTCTCTGCCTT GACTGTG | 189 | 60 |
| *MARVELD1* | GGCCAGCTGTAAGATCATCACA | TCTGATCACAGA CAGAGCACCAT | 100 | 60 |
| *FABP3* | GAACTCGACTCCCAGCTTGAA | AAGCCTAC CACAATCATCGAAG | 103 | 59 |
| *FABP4* | TGGATAGTGCAGCCAGTGTGA | TCCAG TGTGATGCGGTGTGTA | 109 | 60 |
| *SCD* | TCCTGTTGTTGTGCTTCATCC | GGCATAACG GAATAAGGTGGC | 101 | 60 |
| *APOA1* | CGGCGGCTTCTCTTGTATAGC | TTCAA GCGTGAGCTGAAACG | 83 | 60 |
| *LPL* | ACACAGCTGAGGACACTTGCC | GCCATGGATCAC CACAAAGG | 101 | 58 |
| *PPARG* | CCAAATATCGGTGGGAGTCG | ACAGCG AAGGGCTCACTCTC | 101 | 58 |
| *SLC27A1* | GTACCAGCACGAAAGGCTCAA | ATCACAC GGCGCTCTTCAA | 120 | 58 |
| *SLC27A4* | CACGGAGGAACTTCAGATGTGA | GGCCCCGC TATACTGACTATGA | 127 | 59 |

### 2.6. Western Blot Analysis

Total proteins were extracted from MAC-T cells using radio-immunoprecipitation assay (RIPA; Solarbio, Beijing, China) lysis buffer containing 1% phenylmethanesulfonyl fluoride (PMSF; Solarbio, Beijing, China) at 24 h, 48 h and 72 h post-transfection. The protein extracts were quantified using a bicinchoninic acid protein kit (BCA; Solarbio, Beijing, China). Approximately 20 μg of total protein was separated by polyacrylamide gel electrophoresis in a 15% SDS-PAGE gel, and transferred to nitrocellulose membranes at 300 mA, which were then probed with an *SAA1*-specific antibody (ABclonal, Wuhan, China) or a β-actin antibody (ABclonal, Wuhan, China). The membrane was then washed three times with Tris-buffered saline and Tween 20, and then probed with a horseradish peroxidase (HRP)-conjugated secondary antibody (Bioss, Beijing, China) at a 1:5000 dilution. Chemiluminescence detection was performed using a SuperKing™ Hypersensitive luminescent ELC solution (Abbkine, Beijing, China).

### 2.7. Flow Cytometry

An Annexin V/PI Kit (Solarbio, Beijing, China) was applied to detect the apoptosis of MAC-T cells following the manufacturer's instructions. 24 h post-transfection, MAC-T cells were washed with PBS, digested with trypsin and collected by centrifuging at $176 \times g$ at room temperature for 5 min. The cells were re-suspended in binding buffer and stained sequentially with Annexin V and propidium iodide.

### 2.8. Oil Red O Staining

The lipid droplets in MAC-T cells were stained at 72 h post-transfection using an oil red O staining kit (Solarbio, Beijing, China). The MAC-T cells were washed with PBS, and then fixed and stained with oil red O dye for 30 min. After the oil red O dye was washed away with distilled water, the nucleus was stained with a hematoxylin staining solution for 1 min. Finally, the hematoxylin stain was washed off and the MAC-T cells were covered with distilled water. Lipid droplets were observed and photographed under an inverted microscope.

### 2.9. Triglyceride Assay

The levels of cellular TAG in MAC-T cells were measured using a Triglyceride Assay Kit (Applygen, Beijing, China). All the experiments were performed according to the manufacturer's instructions.

### 2.10. Data Analysis

Statistical analyses were performed using the SPSS Statistics 19.0 statistical software (SPSS Inc., Chicago, IL, USA). All the data are expressed as means ± standard errors (SEs). Student's *t*-test was used to determine the statistical significance of the difference between two groups. ImageJ software was used to analyze the relative content of lipid droplets. $p < 0.05$ was considered to indicate statistical significance, and $p < 0.01$ indicated high statistical significance.

## 3. Results

### 3.1. Bta-miR-125a Mimic Suppressed the Gene Expression of SAA1 mRNA by Targeting a Specific Sequence in Its 3′-UTR

We first ran TargetScan (http://www.targetscan.org/vert_71/, accessed on 27 June 2016) and MiRanda (http://www.microrna.org/microrna/home.do, accessed on 27 June 2016), predicting that the 3′-UTR of the *SAA1* mRNA was a target of bta-miR-125a. To validate the regulatory effect of bta-miR-125a on the expression of *SAA1*, we performed a dual-luciferase reporter assay using a plasmid containing the 3′-UTR sequence of *SAA1* mRNA fused to the open reading frame (ORF) of luciferase. The synthetic bta-miR-125a (miR-mimic) was co-transfected with the plasmid into HEK293 cells. The synthetic inhibitor of bta-miR-125a (miR-inhibitor) and a synthetic control miRNA (NC) were also co-transfected with the plasmid. As shown in Figure 2A, at 24 h post-transfection, the luciferase level in the HEK293 cells transfected with miR-125a mimic decreased by 48%, relative to that in cells with the miRNA control ($p < 0.05$). By contrast, the miR-125a inhibitor yielded the same luciferase level as the negative control (Figure 2A). However, when the predicted binding sites of bta-miR-125a were mutated, the luciferase activity was effectively restored to the control level (Figure 2B). These results suggest that bta-miR-125a may inhibit the expression of *SAA1* by targeting its 3′-UTR in MAC-T cells.

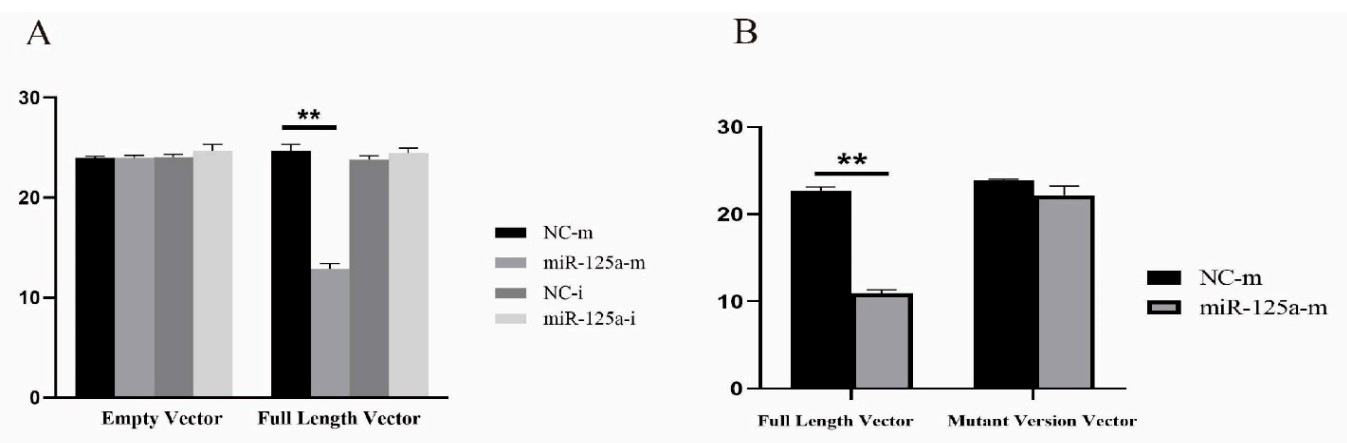

**Figure 2.** MicroRNAs suppressed the expression of *SAA1* by binding to the 3′-UTR target sequence. Luciferase activity in HEK293 cells co-transfected with miRNA mimic, miRNA inhibitor, miRNA control and empty vector for the *SAA1* 3′-UTR. Luciferase activity was assayed 24 h after transfection. All luciferase values were normalized to *Renilla* luciferase. Black columns represent the luciferase activity upon co-transfection with miRNA-mimic control. Middle gray columns represent the luciferase activity upon co-transfection with miRNA-inhibitor control. Dark gray columns represent the luciferase activity upon co-transfection with miRNA mimic. Light gray columns represent the luciferase activity upon co-transfection with miRNA inhibitor. (**A**) represents the luciferase activity of *SAA1* after over- or under-expression of miR-125a compared with controls; (**B**) represents the luciferase activity of *SAA1* after transfecting the mutant vector and miR-125a, compared with control. ** Very significant difference between the control and the treatment ($p < 0.01$).

### 3.2. Bta-miR-125a Inhibits SAA1 Expression in MAC-T Cells

After the transfection of cells with the bta-miR-125a mimic and inhibitor, our flow cytometry findings indicated that there were no obvious differences in the apoptosis rates of MAC-T cells among the groups (Figure 3A).

Our qRT-PCR results showed that the expression of bta-miR-125a was significantly up-regulated in the mimic group, compared with negative-control group (NC; $p < 0.01$), while the opposite trend was found for the expression of bta-miR-125a upon transfection with the inhibitor (Figure 3B). These results indicate that bta-miR-125a was successfully over-expressed and inhibited in the corresponding groups.

In addition, transfection with the miR-125a mimic significantly down-regulated the expression of the *SAA1* gene, compared with that in the NC group ($p < 0.01$); by contrast, transfection with miR-125a inhibitors significantly elevated the expression of the *SAA1* gene, compared with that in the NC-i group ($p < 0.01$); Figure 4A,B.

The results of Western blotting showed that, compared with that in the negative-control group (miR-NC-m), the protein expression of *SAA1* decreased gradually in MAC-T cells transfected with the miR-125a mimic at 48 and 72 h post-transfection, while MAC-T cells transfected with miR-125a inhibitors showed higher levels of the *SAA1* protein than cells in the NC-i group at 48 and 72 h post-transfection (Figure 4C).

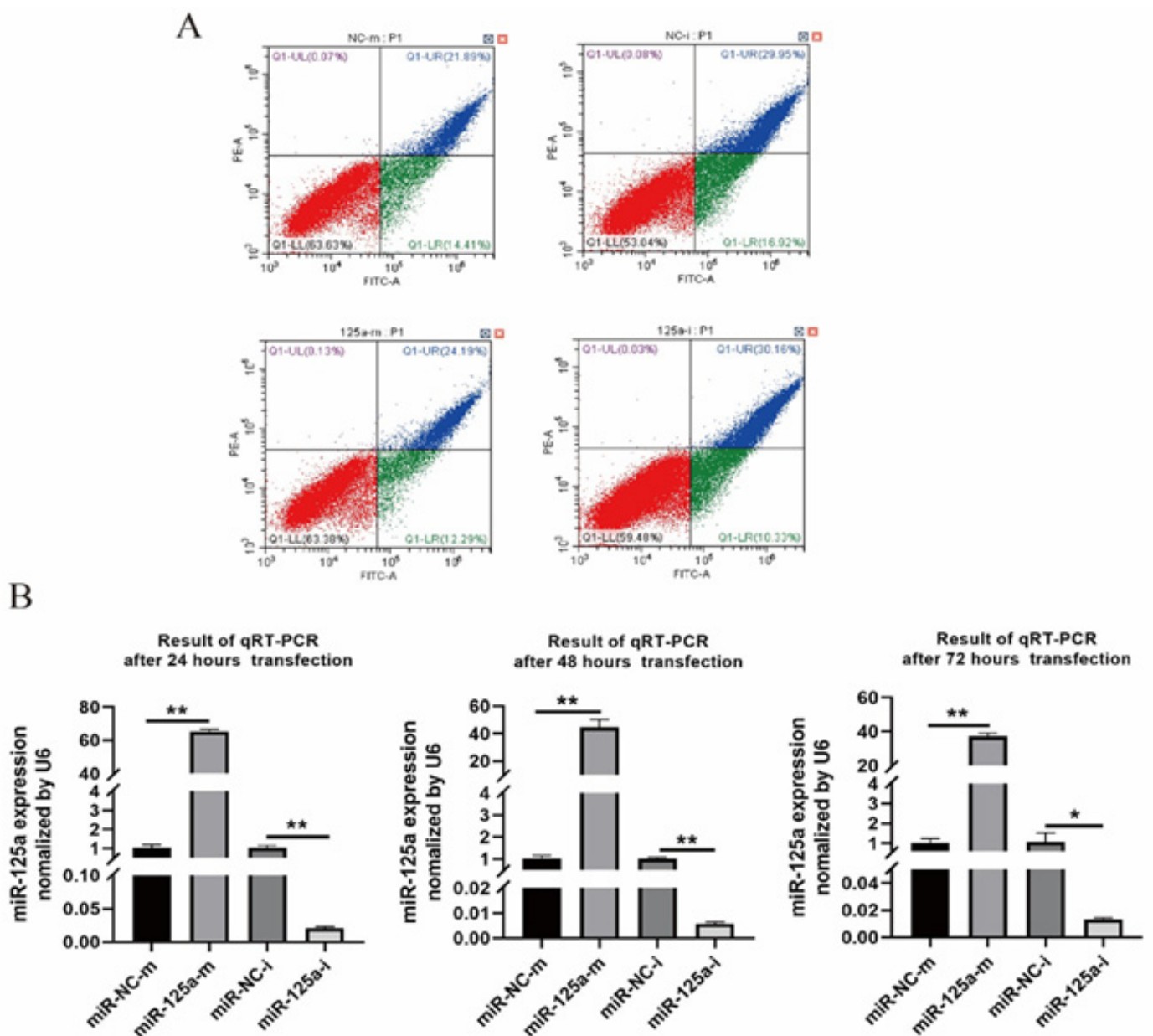

**Figure 3.** (**A**) After transfection with miR-125a mimic and inhibitor for 72 h, flow cytometry was used to detect the apoptosis rate of MAC-T cells; (**B**) Results of qRT-PCR. The expression levels of miRNAs at 24, 48 and 72 h after transfection. * Significant difference between the control and the treatment ($p < 0.05$); ** Very significant difference between the control and the treatment ($p < 0.01$).

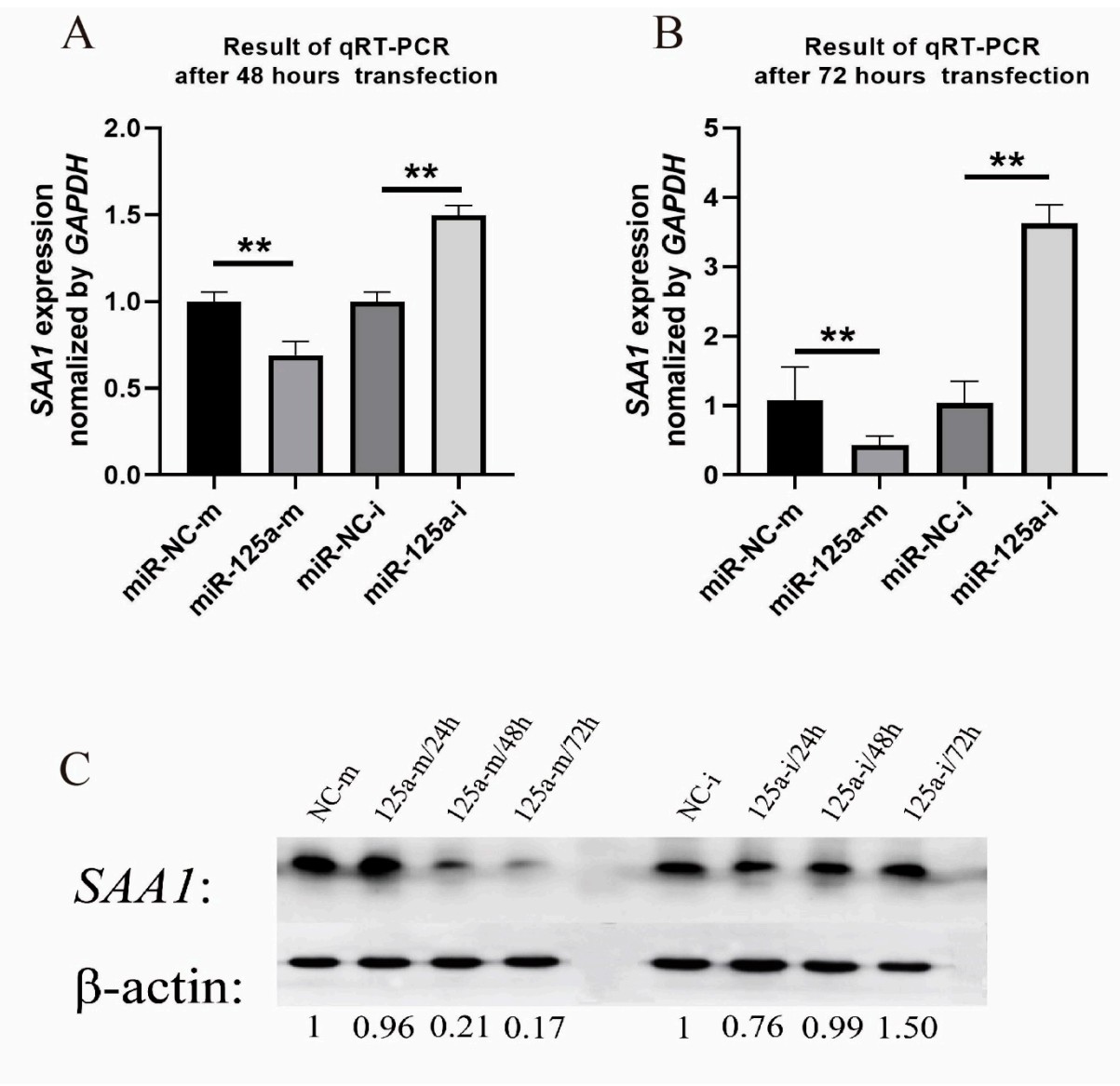

**Figure 4.** MiR-125a regulates *SAA1* expression in dairy cow mammary epithelial (MAC-T) cells. (**A**) Results of qRT-PCR: the expression of *SAA1* at 48 and 72 hours after transfection; (**B**) Western blotting result map and quantification; PhotoShop was used to calculate the gray values; (**C**) Western blot of *SAA1* and their negative controls using β-actin as a reference control. ** Very significant difference between the control and the treatment ($p < 0.01$).

### 3.3. Bta-miR-125a Regulates Expression of Lipid-Related Genes in MAC-T

The real-time qPCR results revealed that, relative to the control, the ectopic over-expression of bta-miR-125a strongly up-regulated the expression of *SLC27A1* ($p < 0.05$), *FABP3* ($p < 0.01$), *FABP4* ($p < 0.01$), *LPL* ($p < 0.01$), *PPARG* ($p < 0.01$) and *SLC27A4* ($p < 0.01$); see Figure 5A. By contrast, cells transfected with the bta-miR-125a inhibitor displayed marked down-regulation of *LPL* ($p < 0.05$), *SLC27A1* ($p < 0.05$) and *PPARG* ($p < 0.05$); see Figure 5B.

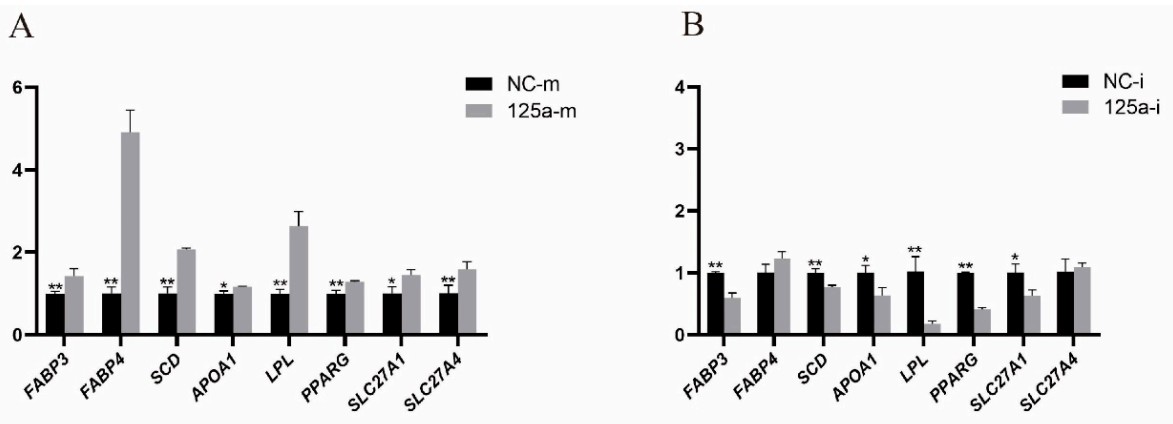

**Figure 5.** (**A**) Expression of lipid-related genes after miR-125a-mimic transfection; (**B**) Expression of lipid-related genes after miR-125a-inhibitor transfection. * Significant difference between the control and the treatment ($p < 0.05$); ** Very significant difference between the control and the treatment ($p < 0.01$).

*3.4. Bta-miR-125a Enhances Lipogenesis in Bovine Mammary Epithelial Cells*

The results of oil red O staining showed that transfection with the miR mimic increased the aggregation of lipid droplets, whereas bta-miR-125a knockdown suppressed the aggregation of lipid droplets (Figure 6A). Using the ImageJ software to analyze the relative content of lipid droplets, we found that the content of lipid droplets in cells transfected with the miR mimic was higher than that in the NC group and in the group transfected with miR inhibitors (Figure 6B). In addition, the triglyceride assay revealed that transfection with miR mimics increased the triglyceride content, while transfection with miR inhibitors had the opposite effect ($p < 0.01$; Figure 6C).

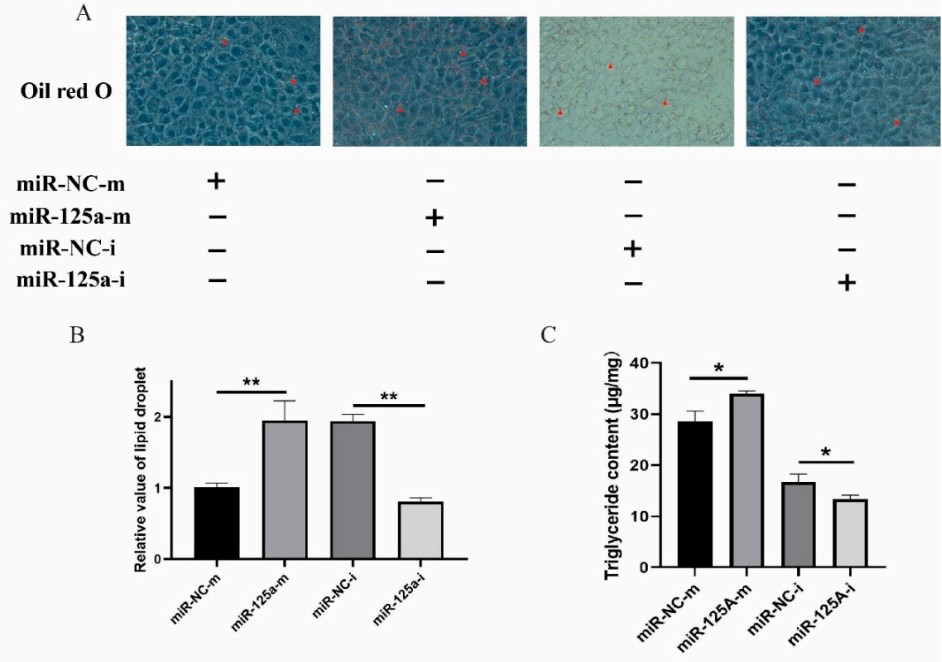

**Figure 6.** (**A**) Oil red O staining: staining at 72 h after transfection, with microscopic examination; (**B**) Oil red O staining: staining at 72 h after transfection, with quantitative analysis of images; (**C**) Triglyceride production by MAC-T cells after miR-125a-mimic and inhibitor transfection for 72 h. All experiments were performed in triplicate. * Significant difference between the control and the treatment ($p < 0.05$); ** Very significant difference between the control and the treatment ($p < 0.01$).

## 4. Discussion

Mammary epithelial cells are the principal milk-producing epithelial cells [30], and are a critical research object for understanding lipid metabolism in dairy cows [31]. Mammary alveolar cell-T (MAC-T) samples were obtained after the transfection of mammary epithelial cells with simian virus-40 (SV-40) large T-antigen, which makes them immortal and stable cells, even after 350 serial passages [32]. Contrary to other established MEC cell lines, MAC-T cells possess the unique characteristic of secreting milk products, and have been widely used in research related to lipid metabolism [33,34]. In the udder, the milk-producing organ in dairy cows, the number of MAC-T cells is considered an essential determinant of milk yield. Therefore, we selected MAC-T cells as an in-vitro model in which to explore milk-fat synthesis. However, compared with cow mammary epithelial primary cells, MAC-T cells have defects, in terms of a relatively insufficient lactation capacity, which we will consider in future research.

Previously, using RNA-seq and small RNA-seq analysis, our group demonstrated that the *SAA1* gene and miR-125a were differentially expressed in the mammary gland between groups of lactating Holstein cows with high and low levels of milk protein and fat percentage [28]. Numerous studies have reported the various effects of the *SAA1* gene in humans [35]; it promotes the release of pro-inflammatory cytokines [36], may induce lipolysis [36,37] and serves as an apolipoprotein of the high-density lipoprotein (HDL) group, which is present in acute-phase serum [38,39]. Previously, we identified *SAA1* as a target for miR-125a [28], which is known to be involved in the differentiation of adipocytes, as well as triglyceride synthesis [40]. In our previous study, *SAA1* and miR-125a were found to be differentially expressed in mammary glands of groups with high and low milk-fat percentages (FP) [28]. Therefore, in this study, we aimed to assess the function of miR-125a and *SAA1* in milk-fat synthesis.

Triglycerides are crucial components of milk fat, representing approximately 98% of the total milk lipids. The content of TAG is an important quality evaluation index for milk [41]. Milk-fat synthesis is affected by multiple physiological, environmental and genetic factors. In cattle, *SAA1* encodes the acute-phase protein serum amyloid A (SAA), which is primarily produced in the liver during the inflammatory response [39]. However, another study has reported a good correlation between amyloid A and mammary inflammation, with a reduced correlation in cows with high SAA [42]. Previous studies have shown that *SAA1* might be involved in mammary gland development through the NF-κB signaling pathway [43], and that *SAA1* over-expression may suppress the growth of mammary epithelial cells [44]. In addition, *SAA1* is a gene essential for milk production in dairy cattle, which negatively regulates milk-fat traits and has been suggested as a genetic marker [12]. In this study, our qRT-PCR and Western blotting results indicate that miR-125a inhibits the expression of *SAA1*. Furthermore, the TAG results reveal that *SAA1* inhibits milk-fat synthesis in MAC-T cells [45]. Therefore, our results demonstrate that *SAA1* may inhibit milk-fat synthesis in MAC-T cells through miR-125a.

MicroRNAs are post-transcriptional regulatory factors that participate in many biological processes, mainly by binding to the 3′ untranslated regions (3′-UTRs) of their target mRNAs and regulating gene expression [46]. Recent studies have reported that miRNAs play an essential regulatory role in milk-fat synthesis. In MAC-T cells, the inhibition of miR-27a-3p, which targets peroxisome proliferator-activated receptor (*PPARG*), may restore the LPS inhibition of milk-fat synthesis. Furthermore, miR-27a-3p inhibition can reverse the LPS-induced down-regulation of *PPARG* expression in LPS-stimulated MAC-T cells [47]. miR-34b has been found to be involved in lipid metabolism, and reduces the accumulation of TAG in primary bovine mammary epithelial cells (BMECs) by targeting lipid-metabolism genes, including *PPARγ*, *C/EBPα*, *FABP4* and *FASN* [48]. Moreover, miR-34b–DCP1A (decapping enzyme 1A) might be an essential axis for milk-fat synthesis in BMECs and the production of beneficial milk components [48]. Furthermore, miR-221 has the ability to inhibit the proliferation of mammary gland epithelial cells by targeting *STAT5a* and *IRS1*, which are considered critical genes in the PI3K–Akt/mTOR and JAK–

STAT signaling pathways, respectively [49]. It has recently been shown that changes in the expression of miR-24 in goat mammary epithelial cells (GMECs) may increase the unsaturated fatty acids and change the TAG content. In ruminant mammary cells, the fatty acid synthase (*FASN*) gene appears to be a direct target of miR-24, where miR-24/*FASN* have exhibited a potential role in controlling lipid metabolism [50]. Moreover, miR-125a-5p was expressed at lower levels in the adipose tissues of mice fed a high-fat diet than in mice fed standard chow. miR-125a-5p expression has also been found to be strongly up-regulated (nearly five-fold), when 3T3-L1 pre-adipocytes were induced to differentiate into mature adipocytes. Functional analysis has indicated that the over-expression of miR-125a-5p promoted the proliferation of 3T3-L1 pre-adipocytes and inhibited their differentiation [51]. Most of the miRNAs in the above-mentioned studies were shown to be involved in milk-fat synthesis. The contrasting results might be attributable to the complex regulatory network of miR-125a, which impairs the expression of genes involved in the lipid synthesis pathway (e.g., *FABP3*, *LPL* and *PPARG*). Our study demonstrated that miR-125a altered lipid accumulation and TAG content, consistent with the results of previous studies on MAC-T and GMECs [52]. The observed increase in cellular TAG was consistent with the increase in milk-fat-related genes (e.g., *PPARG*, *LPL*, *FABP4* and *FABP3*). *PPARG* is a member of the nuclear peroxisome proliferator-activated receptor (PPAR) family, which regulates the transcription of multiple genes and consists of three sub-types: PPAR-α, -β and -γ. *PPARG* promotes cellular processes that involve lipid accumulation [53]. In the bovine mammary gland, the mRNA expression level of *PPARG* during lactation is remarkably changeable, suggesting its prominent role in bovine milk-fat synthesis [54,55]. The increased utilization of fatty acids (FAs) in *PPARG*-knockout mice increased the synthesis of inflammatory lipids; hence, the production of toxic pro-inflammatory milk is related to a lack of TAG synthesis [3]. Data from an IPA analysis, indicating that the over-expression of PPARG down- or up-regulated these upstream transcription factors, further supports our previous hypothesis that PPARG regulates the gene network related to fatty acid metabolism, in either a direct or indirect manner [56]. *SAA1*, *PPARG*, *FABP3*, *FABP4*, *SCD*, *APOA1*, *LPL*, *SLC27A1* and *SLC27A4* are enriched in the PPAR signaling pathway, according to the KOBAS/DAVID software. Furthermore, the up-regulation of *PPARG* expression by *FABP3* significantly promoted the accumulation of lipid droplets [47]. Consequently, our results reveal the ability of miR-125a to control lipid accumulation and TAG content through the expression of *SAA1* and other fat-metabolism-related genes in MAC-T. We speculate that miR-125a promotes the formation of lipid droplets in MAC-T cells by targeting *SAA1*.

In this study, we investigated the miR-125a-based regulatory mechanisms of *SAA1* at the cellular level. Our results indicate that miR-125a promotes the formation of lipid droplets in MAC-T cells by targeting *SAA1*. Overall, the results in this paper show that miR-125a can control the synthesis of milk fat in MAC-T cells by targeting *SAA1*.

## 5. Conclusions

Our previous research revealed the genes related to milk-fat metabolism and their corresponding miRNAs. This study reveals that miR-125a significantly down-regulates the expression of *SAA1* through binding to a specific target sequence in its 3′-UTR. The results indicate that these molecules may play critical roles in the regulation of milk-fat metabolism in dairy cattle. More in-depth investigations are required, in order to validate the biological mechanisms of *SAA1* and miR-125a in the formation of milk production traits in dairy cattle.

**Author Contributions:** X.C., Z.F. and T.Y. performed the RNA-related experiments and data analysis; T.Y. and J.F. helped to interpret the results; and X.C. and C.W. wrote the manuscript and participated in the experimental design. All authors have read and agreed to the published version of the manuscript.

**Funding:** This research was funded by the Scientific and Technological Innovation Programs of Higher Education Institutions in Shanxi, China (Grant No. 2019L0037), and Shanxi Province Science Foundation for Youths, China (Grant No. 201801D221285).

**Institutional Review Board Statement:** Not applicable.

**Informed Consent Statement:** Not applicable.

**Data Availability Statement:** All the relevant data are available within the manuscript.

**Conflicts of Interest:** The authors declare no conflict of interest.

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
