# Peer review of "Bta-miR-125a Regulates Milk-Fat Synthesis by Targeting SAA1 mRNA in Bovine Mammary Epithelial Cells"

_agriculture, doi:10.3390/agriculture12030344_

Round 1

Reviewer 1 Report

Authors improved their manuscript based on comments of all reviewers.

Author Response

In general, I have no comments to the manuscript.

I just suggest to the authors that rewrite the conclusion in order to make them more straightforward with the study and the results here presented.

Response: Thanks very much for the Reviewer’s comments. We have improved the conclusion (Please see Line 403-412).

Reviewer 2 Report

Comments to Bta-miR-125a Regulates Milk-Fat Synthesis by Targeting SAA1 mRNA in Bovine Mammary Epithelial Cells.

In general, I have no comments to the manuscript.

I just suggest to the authors that rewrite the conclusion in order to make them more straightforward with the study and the results here presented.

Author Response

Authors improved their manuscript based on comments of all reviewers.

Response: Thanks very much for the Reviewer’s comments. We have improved the manuscript based on the other review.

This manuscript is a resubmission of an earlier submission. The following is a list of the peer review reports and author responses from that submission.

Round 1

Reviewer 1 Report

The main question addressed by the research is focused on expression of miRNAs and control of certain genes which regulate production of milk fat in dairy cows mammary epithelial cell. The research is relevant and contributes to the deeper knowledge on metabolism of milk fat synthesis in mammary glands.  The topic is original and represents the extension of previous research already done by the same research group. Result from this work as well as results from the previous research contribute to the better understanding of milk fat synthesis in ruminants.  Despite some small technical errors, the paper is written well enough. The text is clear and is easy to read but for those readers that have background in genetics or molecular biology. The conclusions are consistent with the results from the research. However, one or two sentences on practical implication of obtained results from research should be given in conclusion. 

Line 46-47: There are recent findings related to the underlying mechanisms of synthesis of milk fat, such as Chen et al, 2019 (J. Agric. Food Chem. 2019, 67, 14, 3981–3990). It would be wisely to consider this reference for citation and re-write the end of sentence.

Line 233: Term “extreme significance” should be replaced with “high statistical significance”.

Line 346: The comma punctuation mark stands alone at the beginning of the row.

Line 323: The space is needed between word “research” and “[31-32]”. Check for the same missing space in the entire part “4. Discussion”.

Author Response

Dear Editor,

Thank you very much for providing an opportunity for us to revise our paper for your journal. We also wish to thank you and your reviewers for the positive comments and constructive suggestions on the manuscript. We have made the essential revisions suggested by the reviewers as follows.

Responses to Reviewer 1:

The main question addressed by the research is focused on expression of miRNAs and control of certain genes which regulate production of milk fat in dairy cows mammary epithelial cell. The research is relevant and contributes to the deeper knowledge on metabolism of milk fat synthesis in mammary glands.  The topic is original and represents the extension of previous research already done by the same research group. Result from this work as well as results from the previous research contribute to the better understanding of milk fat synthesis in ruminants.  Despite some small technical errors, the paper is written well enough. The text is clear and is easy to read but for those readers that have background in genetics or molecular biology. The conclusions are consistent with the results from the research. However, one or two sentences on practical implication of obtained results from research should be given in conclusion.

Line 46-47: There are recent findings related to the underlying mechanisms of synthesis of milk fat, such as Chen et al, 2019 (J. Agric. Food Chem. 2019, 67, 14, 3981–3990). It would be wisely to consider this reference for citation and re-write the end of sentence.

Response: Thanks very much for the Reviewer’s comments. We totally agree with your comments. Done as requested (Please see Line 46-48).

The citation reference as follows:

Chen Z, Chu S, Wang X, Fan Y, Zhan T, Arbab AAI, Li M, Zhang H, Mao Y, Loor JJ, Yang Z. MicroRNA-106b Regulates Milk Fat Metabolism via ATP Binding Cassette Subfamily A Member 1 (ABCA1) in Bovine Mammary Epithelial Cells. J Agric Food Chem. 2019 Apr 10;67(14):3981-3990. doi: 10.1021/acs.jafc.9b00622. Epub 2019 Mar 28. PMID: 30892026.

Line 233: Term “extreme significance” should be replaced with “high statistical significance”.

Response: Thanks very much for your suggestion. Done as suggested (Please see Line 214-215).

Line 346: The comma punctuation mark stands alone at the beginning of the row.

Response: Thanks very much for your suggestion. Done as suggested (Please see Discussion).

Line 323: The space is needed between word “research” and “[31-32]”. Check for the same missing space in the entire part “4. Discussion”.

Response: Thanks very much for your suggestion. We have checked carefully in the part of Discussion and done as requested.

Reviewer 2 Report

I have raised some points throughout the manuscript and I have noted them using the annotation tools of Adobe acrobat. Some major points are in the statistics section and also in the hypothesis formation and the choice of cell line for the Luciferase sytem transfection. Finally, the manuscript as a whole needs extensive editing of English language and I would suggest to either use a professional service or ask a native-English speaking colleague to edit it

Author Response

Dear Editor,

Thank you very much for providing an opportunity for us to revise our paper for your journal. We also wish to thank you and your reviewers for the positive comments and constructive suggestions on the manuscript. We have made the essential revisions suggested by the reviewers as follows.

Responses to Reviewer 2

T I have raised some points throughout the manuscript and I have noted them using the annotation tools of Adobe acrobat. Some major points are in the statistics section and also in the hypothesis formation and the choice of cell line for the Luciferase sytem transfection. Finally, the manuscript as a whole needs extensive editing of English language and I would suggest to either use a professional service or ask a native-English speaking colleague to edit it.

Response: Thanks very much for the Reviewer’s comments. We totally agree with your comments.

L16 This should be rephrased as it completely disregards milk protein which is at least equally important in terms of nutritional value

L44 replace with triacylglycerides. Although both terms are synonymous and acceptable you are using triacylglycerides in the rest of the manuscript

L74 Mention species

L79 redundant information already covered by previous sentence

L83 nomenclature of genes is not consistent throughout the manuscript

L104 I do not think it is suitable to list the techniques used. You will address thos in the following section

L105 This sentence has no place in the Introduction section. But even as a sentence it is too strong you just revealed a link between one miRNA and one gene.

Response: Thanks very much for your suggestion. Done as requested (Please see Introduction).

L100,110 From what l read in your paper27) there are more associations bet ween miRNAs and protein encoding genes. There is no explanation behind your choice to follow up on this specific pair. Can you please elaborate on this?

Response: Thanks very much for your suggestion. Our initial RNA sequencing (RNA-seq) revealed that the TRIB3/PTHLH/SAA1/SAA3/M-SAA3.2 genes were differentially expressed in the mammary glands of lactating Holstein cows with extremely high versus low phenotypic values of milk protein and fat percentage. To further validate the genetic effect and potential molecular mechanisms of these genes involved in regulating milk production traits in dairy cattle, we performed the study through genotype-phenotype associations. Among the results, seven SNPs (single nucleotide polymorphisms) in the SAA1 gene have been significantly associated with milk yield and composition traits (Yang S, Gao Y, Zhang S, Zhang Q, Sun D. Identification of Genetic Associations and Functional Polymorphisms of SAA1 Gene Affecting Milk Production Traits in Dairy Cattle. PLoS One. 2016;11(9): e0162195. Published 2016 Sep 9. doi:10.1371/journal.pone.0162195). Based on the results of the study through genotype-phenotype associations, we focused on TRIB3/SAA1/SAA3/M-SAA3.2 and the miRNAs. We will do more work on these genes and the miRNAs.

Figure 1 1B does not give the location(position within the utr

sequence should be given eg 535 CUGCCU.....CUCAGGGC 567) as stated, just the sequence. Furthermore, the underlined sequence is not the mutated(deleted) sequence but rather the miR-125a sequence.....1C should be removed it does not offer any useful information

Response: Thanks very much for your suggestion. Done as requested (Please see Fig 1).

L143 Why did you use the luciferase reporter system in HEKs and not on MAC-Ts

Response: Thanks very much for your suggestion. Luciferase assays are quick, highly sensitive, have wide dynamic range, and are cheap to perform. Because of their simplicity and versatility, and because of the absence of endogenous luciferase activity in HEK293T, Hela, et al., this test can be used to study the interaction between miRNA and target gene. Firstly, long fragment plasmids are easy to be transferred into 293T cell lines, however, it is so difficult to be transferred into MAC-T cells. Moreover, more research about luciferase reporter gene system used 293T cell lines. Therefore, we decided to use 293T cell lines for the experiment.

The research as follows:

Deng Z, Zhang Y, Li L, Xie X, Huang J, Zhang M, Ni X, Li X. A dual-luciferase reporter system for characterization of small RNA target genes in both mammalian and insect cells. Insect Sci. 2021 Jul 7. doi: 10.1111/1744-7917.12945. Epub ahead of print. PMID: 34232550.

Huang X, He M, Huang S, et al. Circular RNA circERBB2 promotes gallbladder cancer progression by regulating PA2G4-dependent rDNA transcription[J]. Molecular cancer, 2019, 18(1): 166.

Yin Q, Wang J, Fu Q, et al. CircRUNX2 through has‐miR‐203 regulates RUNX2 to prevent osteoporosis[J]. Journal of cellular and molecular medicine, 2018, 22(12): 6112-6121.

Luo E, Wang D, Yan G, et al. The NF-κB/miR-425-5p/MCT4 axis: A novel insight into diabetes-induced endothelial dysfunction[J]. Molecular and cellular endocrinology, 2019: 110641.

Wu Z, Huang W, Wang X, et al. Circular RNA CEP128 acts as a sponge of miR-145-5p in promoting the bladder cancer progression via regulating SOX11[J]. Molecular Medicine, 2018, 24(1): 40.

L154 I suppose the authors mean reactions

Response: Thanks very much for your suggestion. We did three independent experiments. 30 ng vector with 50 μl opti-MEM and 30 nM mimic miRNA were co-transfected into three wells with 1 μl Lipofectamine 2000, and then relative firefly luciferase activities were measured three times after transfection with the Dual-Luciferase Reporter Assay Kit for every well. We compared the three results, and then we carried out the next experiment when the results were consistent. The same methods were used by inhibitor miRNA, NC group.

L172 That means that you omitted samples due to low quality?

Response: Thanks very much for your suggestion. That means that all the RNAs with optical density between 1.8 to 2.0 at 260/280 nm values. Otherwise, we will re-extract RNA and carry out the next experiment until the RNA quality meets the conditions.

L181 which one did you actually use?

Response: Thanks very much for your suggestion. The expression levels of the target gene SAA1 and the Lipid-Related Genes SLC27A1, FABP3, FABP4, LPL, PPARG, SLC27A4 in the PPAR signaling pathway were quantified using GAPDH as a reference gene. When the expression levels of bta-miR-125a were performed by qRT-PCR and normalized against the U6 for each sample.

L184 What do you mean that the results are representative? Did you use all the data obtained from the independent experiments? How many biological replicates did you use? How many technical replicates?

Response: Thanks very much for your suggestion. We did three independent experiments using cells and then qRT-PCR was carried out by three replicates for every experiment. That means we used three biological replicates. For every biological replicate, we used three technical replicates.

Table1 This table does not make sense. What are the RT primers? Why are there no sequences in U6?

Response: Thanks very much for your suggestion. RT primers were used when total RNA were reverse-transcribed to cDNA miRcutePlus miRNA First-Strand cDNA Kit (Tiangen, Beijing, China). The U6 primers were designed by the company named Tiangen and they will provide us the sequence.

Table 2   Why did you use a pair of primers amplifying a 2433 region of the transcript?

Response: Thanks very much for your suggestion. The 3′un-translated region (UTR) of SAA1 that contains the miRNAs target sites in the 3’UTR was PCR amplified using DNA collected from the bovine mammary gland samples.

L216 unneeded repetition you could just add the 72h post-transfection info in the previous sentence

Response: Thanks very much for your suggestion. Done as your suggestion (Please see 2.8 ).

L232 the t-test is not appropriate for some of your comparisons as there are more than 2 groups (FIG2A: NC-m, miR-125a-m, NC-i, miR-125-a-i

Response: Thanks very much for your suggestion. We focused on the comparisons of NC-m and miR-125a-m, NC-i and miR-125a-i, respectively.

L233 I find this sentence unacceptable. A difference is either significant or non-significant. It should also be changed to "The significance level was set to .05" or "P values below 0.05 were considered to be statistically significant" If the authors would like to be more strict they can set the significance level lower

L247 I suppose that the inhibitor was co-transfected in the presence of miR-125a. If that is the case please rephrase to state that.

Response: Thanks very much for your suggestion. We have rephrased this part as one review’s suggestion.

Figure 2 The figure legend should only contain the necessary information for the reader to interpret the image. Therefore the phrase " microRNAs suppress the..." should be removed."

Luciferase activiy was assayed..." should also be removed as it is already described in M&M

section. All abbreviations like " NC-m, miR-125... etc should be explained in the legend and all descriptive text such as "dark

Figure 3B It would help for visualization if you could divide your y-axis into two sections (0-2 and 20-70 or80). Also in the first graph there is no SE. Why is that.

Response: Thanks very much for your suggestion. Done as your suggestion (Please see Fig 2 and Fig 3).

Figure 4C What software did you use to calculate intensity values of the bands?

Response: Thanks very much for your suggestion. To calculate intensity values of the band, Adobe Photoshop 2020 were used.

Figure 6A   How many photographs were used per treatment from the same well?

Response: Thanks very much for your suggestion. At least 5 photographs were used per treatment.

L318-325   This passage belongs in the introduction section

L335-339   this is just a repetition from the introduction

L343-348   This passage does not offer anything to the discussion and should be omitted

L356    this is not accurate. Although most common in the 3'UTR miRNA binding sites can be found in the 5'UTR or even in the CDS (DOI 10.1186/s12943-018-0765-5)

L409 That is a very strong sentence please rephrase

L416 Please rephrase to include only the trait that these particular molecules can influence

Response: Thanks very much for your suggestion. We have rephrased the manuscript using the editing services (https://www.mdpi.com/authors/english) (Please see Discussion).

Finally, we wish to thank you and your reviewers again for the valuable suggestions. If you have any question on this manuscript, please feel free to contact me. Thank you again for your time and favorable consideration.

Sincerely

Changxin Wu, PhD, Professor

Xiaogang Cui, PhD

Email:[email protected]

Reviewer 3 Report

The document is adequately written. The problem is appropriately described and the materials used are well described. The description of the statistical analysis should be improved as it lacks details of how it was performed, the discussion is extensive and detailed. The conclusions should be improved by omitting results from previous studies.

Author Response

Dear Editor,

Thank you very much for providing an opportunity for us to revise our paper for your journal. We also wish to thank you and your reviewers for the positive comments and constructive suggestions on the manuscript. We have made the essential revisions suggested by the reviewers as follows.

Responses to Reviewer 3

Comments and Suggestions for Authors

The document is adequately written. The problem is appropriately described and the materials used are well described. The description of the statistical analysis should be improved as it lacks details of how it was performed, the discussion is extensive and detailed. The conclusions should be improved by omitting results from previous studies.

Response: Thanks very much for the Reviewer’s comments. Done as suggested.

Reviewer 4 Report

Authors try to elucidate the connection between Bta-miR-125and  Fat Synthesis  in Bovine Mammary Epithelial Cells. Their results provide valuable novel information about the role of Bta-miR-125 in the mammary gland of bovine species. There are some points that could be improved in order the article to be considered.

Title. Please avoid only abbreviations support them with full names (SAA1). Consider also to add in the title " Targeting SAA1 gene or mRNA"

Abstract. Add full names the first time you refer an abbreviation (i.e. MAC-T)

Introduction

l 51-55 Add briefly the role of the other two isoforms (3+4)

l 83 Elovl5 gene --add full name

Materials and Methods

l 156 & 185 what do you mean three independent experiment? three trials (replicates) or three experiments using cells from the the begining of DNA experiments? Please explain.

l 186 Table 1 and 2 can be merged

Figure 4 capture for C image is missing. B capture possibly correspond to C image

l 307 Provide analytic information how you used Imaje J software in the materials and methods section

l 309 How you determined the content? please explain

l 311. Please explain how you determined triglyceraide content. Chemically and with which method?

Discussion

l. 317 please rephrase

l 318 for understanding mammary  lipid metabolism

Is there any conection of the miR-125with NADPH depented dehydrogenases that involved in the fatty acid synthesis in ruminants in a major extent. For instance G6PD attributes almost the 50-60% of the required energy by terms of NADPH for de novo fatty acid synthesis. A major contribution has also ICDH. It would be interesting if there would be any sign of connection. 

Additional files
Additional files are missing for further evaluation

Author Response

Dear Editor,

Thank you very much for providing an opportunity for us to revise our paper for your journal. We also wish to thank you and your reviewers for the positive comments and constructive suggestions on the manuscript. We have made the essential revisions suggested by the reviewers as follows.

Responses to Reviewer 4

Authors try to elucidate the connection between Bta-miR-125 and Fat Synthesis in Bovine Mammary Epithelial Cells. Their results provide valuable novel information about the role of Bta-miR-125 in the mammary gland of bovine species. There are some points that could be improved in order the article to be considered.

Title. Please avoid only abbreviations support them with full names (SAA1). Consider also to add in the title " Targeting SAA1 gene or mRNA"

Response: Thanks very much for your suggestion. Done as suggested.

Abstract. Add full names the first time you refer an abbreviation (i.e. MAC-T)

Response: Thanks very much for your suggestion. Done as suggested.

Introduction

l 51-55 Add briefly the role of the other two isoforms (3+4)

l 83 Elovl5 gene --add full name

Response: Thanks very much for your suggestion. Done as suggested (Please see Line ).

Materials and Methods

l 156 & 185 what do you mean three independent experiment? three trials (replicates) or three experiments using cells from the beginning of DNA experiments? Please explain.

Response: Thanks very much for your suggestion.

1.Luciferase reporter assays: 30 ng vector with 50 μl opti-MEM and 30 nM mimic miRNA were co-transfected into three wells with 1 μl Lipofectamine 2000, and then relative firefly luciferase activities were measured three times after transfection with the Dual-Luciferase Reporter Assay Kit for every well. We compared the three results, and then we carried out the next experiment when the results were consistent. The same methods were used by inhibitor miRNA, NC group.

2.qRT-PCR: Similar with Luciferase reporter assays, we did three experiments using cells and then qRT-PCR was carried out by three replicates for every experiment.

l 186 Table 1 and 2 can be merged.

Response: Thanks very much for your suggestion. We totally agree with your comments, however, we thought that the table will appear messy when Table 1 and 2 were merged.

Figure 4 capture for C image is missing. B capture possibly correspond to C image

l 307 Provide analytic information how you used Imaje J software in the materials and methods section

l 309 How you determined the content? please explain

Response: Thanks very much for your suggestion. Done as suggested (Please see Line ). We counted the lipid droplets in each group by ImageJ software, and then compared the statistical results of miR mimics group with NC group. We found that the content of lipid droplets in cells transfected with the miR mimic was higher than that in the NC group and in the group transfected with miR inhibitors

l 311. Please explain how you determined triglyceraide content. Chemically and with which method?

Response: Thanks very much for your suggestion. According to the manufacturer's protocol, absorbance was detected at 570 nm and quantified using a spectrophotometric microplate reader. This kit can determine the content of triglyceride using glycerol phosphate oxidase method and classic GPO Trinder enzymatic reaction principle. Meanwhile, the kit also follows the clinical test standard of triglyceride stipulated by the World Health Organization (WHO), FDA and China's national code of clinical test operation. It has high sensitivity and the detection range is 20-2000 µmol / L.

Discussion

  1. 317 please rephrase

l 318 for understanding mammary lipid metabolism

Is there any conection of the miR-125 with NADPH depented dehydrogenases that involved in the fatty acid synthesis in ruminants in a major extent. For instance G6PD attributes almost the 50-60% of the required energy by terms of NADPH for de novo fatty acid synthesis. A major contribution has also ICDH. It would be interesting if there would be any sign of connection.

Response: Thanks very much for your suggestion. We have rephrased the manuscript using the editing services (https://www.mdpi.com/authors/english). We consulted a large number of literatures, however, there is no connection of miR-125a and NADPH. Anyway, we thought your suggestion is very interesting and we will do some work in this aspect in our next work. Your suggestion plays an important role in our future research.

Finally, we wish to thank you and your reviewers again for the valuable suggestions. If you have any question on this manuscript, please feel free to contact me. Thank you again for your time and favorable consideration.

Sincerely

Changxin Wu, PhD, Professor

Xiaogang Cui, PhD

Email:[email protected]
